# Health of Polo Horses

**DOI:** 10.3390/ani14121735

**Published:** 2024-06-08

**Authors:** Anton Schumacher, Heidrun Gehlen

**Affiliations:** Equine Clinic, Veterinary Department, Freie Universitaet Berlin, 14163 Berlin, Germany

**Keywords:** polo, horse, sport, injury

## Abstract

**Simple Summary:**

The sport of polo poses a variety of health risks to polo horses, including lameness injury, joint disease, back problems, exercise-induced pulmonary hemorrhage, and infectious disease. This analysis shows that polo horses are exposed to an increased risk of injury and disease due to the demands of the sport, with aspects such as ground conditions, training, and genetic factors playing a role. Prevention is the top priority in this growing sport. Both the associations and the players must work continuously to protect the health of horses in polo and minimize injuries.

**Abstract:**

This literature review analyzes the historical development of polo, its organizational structure, the course of the game, as well as the breeding, rearing, and training practices of polo horses. Frequently occurring ailments, such as musculoskeletal injuries, respiratory diseases, and internal illnesses, are highlighted. Lameness is a major problem, with injuries to the superficial digital flexor tendon being the most common cause. Other notable diseases include exercise-induced pulmonary hemorrhage (EIPH), myositis, rhabdomyolysis, and equine infectious anemia (EIA). To ensure the welfare of polo horses, effective prevention and management strategies are crucial. These include proper training, the adaptation of the ground surface, appropriate shoeing, and compliance with animal welfare guidelines and association rules. Collaboration between associations, players, organizers, and veterinarians is crucial. Promoting responsible management practices and raising awareness among stakeholders can help ensure that polo continues to thrive while maintaining high animal welfare standards.

## 1. Introduction

The sport of polo is growing in popularity around the world and is currently played in approximately 80 countries [1]. Polo is a competitive sport that relies on skill, speed, strategy, and teamwork. With increasing interest, animal health and welfare issues are also coming to the forefront. The primary objective of this review is to provide a comprehensive overview of the existing literature related to animal health in polo. Special emphasis is placed on the most common diseases of polo horses.

Through a systematic analysis of the relevant literature, information on the health aspects of polo horses is collected and processed. In particular, the identified diseases and possible correlations with training, protective measures, and environmental influences on the horses are investigated. These horses are athletes and must meet certain standards in order to excel in this sport. Speed, agility, endurance, and responsiveness are essential qualities required of polo horses [2].

To understand the demands placed on these horses, it is important to understand the dynamics of the sport. In polo, two teams of four players typically compete to score goals on a 150 by 275 m field [3]. While grass is still the traditional playing surface, variations such as sand and indoor polo have become more popular in recent years. A game consists of four to eight chukkas, each lasting a maximum of 7 min and 30 s of pure playtime, with the number of goals scored determining the winner [4]. A chukka in polo can last from approximately 8 to 12 min. Because the average speed and exertion of the game varies due to time interruptions caused by stoppages, such as changing horses, etc., players and ponies have different opportunities for physiological recovery during each chukka. Indoor polo is very popular because it requires fewer chukkas, horses, and players than the outdoor game. This makes it an ideal choice for smaller polo clubs [5]. At the same time, they require players to be more agile to maintain the intensity of the game in a smaller area [6].

The sport’s handicap system, which ranges from −2 to +10, reflects the individual skills of players and determines team handicaps that affect the level of competition [7]. In order to assess the health and well-being of the horses, it is essential to understand the various demands placed upon them by the sport of polo and the different dynamics of the game at different levels of play. Blood chemistry, which adapts to exercise, can be used as an aid in this process [8]. Furthermore, a higher level of play also requires a higher intensity of activity from the horses [1,9]. It is important to consider the role of management during exercise, as adapted playing strategies have the potential to enhance cardiovascular recovery [10].

Therefore, the aim of this review is to examine the existing literature on animal health in polo, focusing on the most common veterinary factors to support future research and recommendations to improve the welfare and performance of polo horses.

To understand the sport of polo, it is important to examine its history. According to reports, in the earliest history of polo, the Turkmen won the first tournament against the Persians. Historical records indicate that the origins of polo date back to the 6th century BC, making polo one of the oldest known equestrian sports in the world [11]. Throughout its history, polo has spread from Asia Minor to India and Europe [1]. The word “polo” comes from the Persian word “Pulu”, which means “ball” [1]. The Calcutta Polo Club, located in Calcutta, India, was founded in 1862 by the colonial British and is considered to be the oldest polo club in the world currently in existence [12].

In the 19th century, polo continued to grow in popularity and eventually became a national sport in Great Britain. The modern rules and regulations of polo were established in 1875 by the newly formed Hurlingham Polo Committee. As a pioneering association, it founded the first polo club in Europe in 1878 and drew up a set of rules for the sport that is still recognized today. Since 1925, the association has continued to exist under the name “Hurlingham Polo Association” with 2000 members in Great Britain and is still responsible for the regulation of polo [13].

The characteristics required for polo horses have evolved over time. Initially, small ponies were common, but as polo mallets and striking techniques improved, larger and faster horses became more desirable. However, the HPA’s attempt to limit the height of the horses failed [2]. The evolution of polo throughout the years demonstrates the significant growth of the sport and the enduring presence of numerous traditions within it.

The sport of polo is closely networked globally through various organizations. The overarching organization in this context is the Federation of International Polo (FIP), which was founded in 1987. As an organization officially recognized by the International Olympic Committee, the FIP includes federations from 82 nations [1].

Argentina is considered the leading nation in polo, followed by the United States and Great Britain. In terms of both horse and player numbers, these nations have a clear lead over the others. The associations in these countries are the Asociación Argentina de Polo, the United States Polo Association, and the Hurlingham Polo Association. They are largely responsible for the organization, rules, and controls. They are authorized to enforce disciplinary proceedings against players and, therefore, play a major role in the area of animal health in the sport of polo. In Germany, the German Polo Association (Deutscher Polo Verband e.V. Munich, Germany) takes over these tasks. With currently around 400 registered polo players, the association represents an up-and-coming polo nation [14].

Ideal polo horses are characterized by their speed, agility, and endurance. In addition, they have a quick-reacting and intuitive nature [2]. There are no restrictions on breed and size according to the rules [15]. Polo horses of the German polo horse breed (Deutsches Polopferd) have been certified, according to the Schleswig-Holstein/Hamburg Studbook, to have an optimal height of about 156 cm [16].

The main requirements for the health of polo horses are robust health, good physical and mental resilience, good fertility, and freedom from hereditary diseases [16]. Cloning in relation to the breeding of polo horses is also being focused on abroad. Adolfo Cambiasso, the best polo player in the world, is seen as a pioneer, having won the world’s most important polo tournament in 2013 with the help of six cloned polo horses [17]. In Argentina, among other countries, cloning is also being focused on in relation to the breeding of polo horses [18]. In Germany, however, cloning is not permitted in the breeding program, and clones and their offspring may not be entered into the studbook [16].

Gender is also playing an increasingly important role in the breeding of polo horses. An Argentinian study from 2013 found that a preimplantation genetic diagnosis (PGD) can be successfully used for sex determination, which is also cited as a success in the context of polo horse breeding. Mares are preferred because they are considered to be easier to train and more agile [19]. Although some sources emphasize the benefits of female horses in polo horse breeding, there is no scientific evidence to support this. In other areas of equestrian sports, the advantage cannot be shared. An English study from 2018 even showed the opposite, namely that stallions and geldings performed better than mares in most event disciplines [20]. The comparison should be viewed with caution because the performance requirements for polo horses do not require the same attributes as those for event horses.

Comprehensive physical preparation is an important part of raising and training polo horses to prepare horses in terms of their health. Training usually begins between the ages of three and four. Some particularly talented horses take part in high-goal competitions as early as the age of six. In the course of time, around the age of 12 to 14, the speed often decreases, and these horses are then used in lower divisions [21]. A study from 2016 showed that training intensity has a direct influence on the condition of endurance polo horses, with the recovery rate after high-speed training being an important indicator [8]. Preseason training was of high importance for more than 98% of polo players interviewed in a survey [22]. In addition, the involvement of experienced riders is very important in the training process [23]. Training games are also used to improve the physical preparation of the horses because the game is distinguished by a high degree of variability [3]. In contrast to most other equestrian sports, the physiological demands of a game are associated with stopping and strong changes in the speed and direction of the movements [3,11,24]. In training games, the interaction between the player and the horse is also trained [23].

Polo horses in training or in regular competition show biochemical variations in lactate, aspartate aminotransferase (AST), alkaline phosphatase (ALP), gamma-glutamyltransferase (GGT), urea, creatinine, phosphorus, and potassium. These results, shown in Table 1., suggest that these biochemical variations are due to the physical condition of the animals [8]. 

The most common training program for polo horses is endurance training. This program enhances the oxidative capacity of the muscles, which is crucial for the horses’ endurance. In addition to increasing oxidative capacity, training also tends to increase glycolytic capacity. Glycolytic capacity is important for short-term intense activity, but there is a potential risk of metabolic disorders such as exercise-induced rhabdomyolysis [25]. The muscles of equines are more reliant on glycolytic rather than oxidative activity. Glycolysis is a process by which glucose and glycogen are broken down into ATP energy without the presence of oxygen. This process is particularly important during short bursts of intense activity in the absence of a sufficient oxygen supply.

Rapid bursts of activity necessitate glycolytic metabolism to meet the sudden energy demands that arise in such instances. Equine musculature is characterized by the presence of numerous fast-twitch glycolytic fibers, which facilitate rapid and powerful contractions. However, these fibers are more susceptible to fatigue than oxidative fibers. 

Glycolytic exercise results in the production of lactic acid, a byproduct of anaerobic metabolism that can cause muscle damage and inflammation. Rhabdomyolysis is defined as the breakdown of muscle fibers and the subsequent release of enzymes and proteins into the bloodstream [26,27].

Blood lactate concentrations in relation to exercise are also used in biochemical studies of polo horses [24,28]. Before training, plasma lactate concentrations show no differences between well-trained and untrained horses. However, plasma lactate is significantly lower in well-trained polo horses during and after training [29]. Other metabolites, such as aminotransferase or creatinine, are also markers of muscle and kidney damage. This also emphasizes that adaptation to exercise is a biologically demanding and active process. Therefore, repeated and measured studies should be performed to assess the consistency of values in response in polo horses. In conclusion, the literature shows that the training of polo horses is a complex matter that requires individual adaptation to the horse’s needs. It is important to keep an eye on both the physical fitness and the biochemical aspects to ensure the health and performance of polo horses. 

A polo match demands different physiological requirements from the horses and riders. The intensity of the game is variable and subject to breaks, which means that the physiological demands are not continuous. In polo, there are significant differences between open polo and women’s polo in many spatial and temporal characteristics, and these differences vary in magnitude from small to very large. For example, the absolute top speed in open polo is 61.5 km/h, while it is 59 km/h in women’s polo.

There is also a large difference in the total distance traveled per chukka: in open polo, players cover significantly more distance per chukka than in women’s polo.

The speed and intensity of the game are also different. In men’s open polo, both the result of the chukka and the distance covered in the match influence the distance covered, with an average of 429.0 m more being covered per chukka in the won matches. In women’s polo, there are no significant differences in the total distance per chukka in relation to the match or chukka result [6,11,30]. 

The warm-up of polo horses before the game is usually short but intense, lasting 3–5 min. Interestingly, this short warm-up period has positive effects on the polo horse’s ability to achieve a higher maximum oxygen uptake during the game [9,31]. A polo horse can be used for two chukkas per match, but it needs at least one chukka of recovery time in between. The primary responsibility for the health and welfare of the horses lies with the player, although the referee will also intervene, especially if there are signs of exhaustion, injury, or stress in the animals [7,23]. A polo horse gallops between 2500 and 3000 m during a chukka. The distance in meters covered in a chukka increases due to a higher overall handicap of the team, and the average speed in km/h is also higher over the duration of the game. As the level of play increases, the demands on the horse’s cardiovascular system and anaerobic metabolism increase accordingly [3,11]. Acute, intense stress during polo tournaments can lead to significant, albeit temporary, physiological changes. For example, a polo horse’s heart rate can reach 210–220 beats per minute during an intense match. Despite these changes, most parameters return to the physiological range immediately after exercise [32]. However, studies have also found that polo can lead to cardiac fatigue in horses. The exact causes and long-term effects of this cardiac fatigue have not yet been conclusively clarified and require further research [33].

The game of polo also involves significant cardiovascular exertion for the player, with the maximum heart rate of players rising to 165 beats per minute in most matches and occasionally exceeding 200 beats per minute. A 2020 study found that if the player’s cardiovascular condition is not sufficient to meet the demands of polo, the joint performance of the player–horse combination may be limited [5].

## 2. Materials and Methods

The systematic literature search was conducted up to February 2023 using the following databases: the Primo University Library database, PubMed, and Google Scholar. For the literature search, the keywords “Polo” and “Horse” were used in combination with “Sport” or “Injury” in the Boolean logic for the query. In our search queries, we utilized the Boolean comparison operators AND, OR, and NOT. All titles were screened, and publications that were not considered relevant after screening were excluded from the results. Studies whose texts referred to polo or polo horses were identified as an essential element. To determine whether the publications were relevant to the present work, the key aspects and conclusions of each study were read to ensure the comprehensiveness of the underlying research question. When analyzing the health status of polo horses, no inclusion or exclusion criteria were used for the type of studies and documents, as only a few sources were available. 

Additionally, we conducted systematic research on the internet and collected data and information from both the Google search engine and the websites of national and international associations. We recognized that non-peer-reviewed sources might have an increased potential for bias, but they can still contribute to enriching the discussion. Despite the limited availability of publications, a high level of quality could be achieved by primarily utilizing scientific texts. Our research primarily focused on German and English literature.

## 3. Results

### 3.1. Health Challenges of Polo Horses

Polo horses are known for their speed, agility, and endurance and are exposed to constant stress during play. This constant strain increases the risk of sport-related illnesses and injuries, especially lameness. An intensive analysis of various sources showed that polo horses frequently suffer from recurring illnesses. An in-depth analysis of multiple sources revealed that polo horses often suffer from recurrent illnesses, with the risks increasing with the length of exposure and the number and duration of competitions. A 2015 study found that 10.6% of polo horses suffered from sport-related injuries [22]. There is also a significant association between the duration of polo horse ownership and the risk of injury. Longer periods of ownership are associated with a reduced risk of injury [3]. 

The main factor that players cite as an important risk for injury to polo horses is hard ground [22]. A study not only focusing on polo horses showed that the interaction between the horse’s hoof and the ground plays a crucial role in determining the load on the horse’s extremities, which in turn influences injury and performance [34]. As horses have a limited ability to adapt their gait to surface conditions, veterinarians, in collaboration with farriers, can significantly influence this interaction by adjusting the ground cover and shoeing strategies. This can help to reduce the risk of injury and improve the performance of polo horses [34].

A study conducted with a very small number of subjects (60 horses) indicated that polo horses demonstrate marked movement asymmetry when traversing a straight line on a hard surface when in a trot. More than half of them could have been considered clinically lame. The asymmetries did not appear to be related to the age of the horses. Polo horses, in particular, predominantly canter, but the study referred to polo horses that mostly trotted [35]. Intensive braking maneuvers, especially when accompanied by turns, expose the joints and tendons of the animals to eccentric stresses. This makes them particularly susceptible to musculoskeletal injuries [36]. In fact, such injuries are the most commonly reported injuries in polo horses [22]. In particular, injuries to the superficial digital flexor tendon are the most common [21].

#### 3.1.1. The Superficial Digital Flexor Tendon 

The superficial digital flexor is the most frequently injured structure of the musculoskeletal system in polo horses and often leads to premature retirement. Using high-speed biplanar fluoroscopic cinematography, it was shown that tendon tissue, even when visually healthy, is often not fully healed when trotting [37]. An ultrasonography study of 40 polo horses also showed that almost half of the horses had ultrasonographic changes in the superficial flexor tendon [38]. Stress-related ligament and tendon injuries in polo occur not only in the superficial flexor tendon but also in the collateral ligaments of the distal interphalangeal joints. Polo horses over nine years of age have an increased risk of collateral ligament disease of the distal interphalangeal joints [39]. Joint diseases, especially osteoarthritis, are also a common cause of lameness in polo horses. Osteoarthritis of the metacarpophalangeal joint has been identified as the main cause of early retirement. The concentration of chondroitin sulfate in the synovial fluid appears to be an early marker of articular cartilage damage. A study from 2014 showed that polo horses with increased chondroitin sulfate concentrations between 200 and 150 µg/mL later developed osteoarthritis [40].

The treatment of lameness and injuries in polo horses varies depending on the severity and cause. Common approaches include hydrotherapy, cold applications, anti-inflammatory medications, and local injections. Further research is being conducted to investigate the potential benefits of bone marrow-derived mesenchymal stem cells at a concentration of 5 × 10⁶ cells/mL when administered into the core lesion under ultrasound guidance for the treatment of tendon injuries in horses [41]. According to Crowe et al., shockwave therapy may also improve the chance of recovery [42].

#### 3.1.2. Trauma in the Polo Horse

In the case of trauma as a health challenge, we focus here on the physical injuries to polo horses. These include injuries caused by external forces such as accidents, falls, or blows. It is important to differentiate that acute trauma is a sudden and severe injury caused by a single event, while chronic trauma is caused by repeated or sustained stress or injury over a longer period.

Traumatic injuries are not uncommon in polo horses and are frequently reported [22]. However, not only in polo but also in general horse populations, a study showed that the prevalence of traumatic injuries is approximately 25%, much higher than the 7.7% reported in another study [43]. Recent research has also considered injuries that may not have required veterinary care [44].

An exemplary case study on the extent of injuries in polo describes how a seven-year-old polo horse was hit in the left knee joint by its own rider’s polo mallet during a match. This trauma resulted in pronounced lameness [45].

The literature also contains information on ophthalmic abnormalities that may be due to previous trauma. These anomalies are described with a prevalence of 9.2%, with polo horses in particular, but also polocrosse horses, being affected [46]. Polocrosse is an equestrian sport that combines elements of polo and lacrosse. It should be emphasized that it is against the rules of the sport to play a polo horse that is “blind in one eye”.

Improperly fitted bits and bridles can cause oral trauma, such as mandibular bone spurs, commissural ulcers, and tongue injuries in polo horses and racehorses. A study examined the most common types of oral injuries in polo horses and racehorses that could have been caused by improperly fitted bits and bridles. Racehorses showed a significantly higher severity and frequency of oral injuries compared to polo horses. Tongue injuries were only observed in polo horses. Racehorses had more severe injuries to the lip commissures and bone spurs. In polo horses, injuries occurred over a longer period. Thus, most oral injuries from bits and bridles are chronic over time [47].

#### 3.1.3. Back Disorder

Back problems in polo horses can cause the impairment of the musculoskeletal system. The integrity and functionality of the back are essential for sport horses. A study of 181 horses showed that numerous factors could contribute to the development of back problems, some of which were specifically described in the scientific literature. The horses in the study were mainly used for show jumping and dressage. The various horse breeds that appeared to be predisposed to back problems included thoroughbreds, Arabians, polo horses, and warmbloods. A gender-specific predisposition for back problems in horses is not yet known. However, the literature shows that horses from the age of five are particularly susceptible to such disorders [48]. Equine back disorders (EBDs) are divided into primary and secondary disorders. The most frequently occurring disorders are soft tissue disorders, such as muscle soreness and swollen muscles, where the longissimus muscle and the supraspinous and dorsal sacroiliac ligaments are frequently injured. Particularly problematic primary EBDs include vertebral diseases, the type and frequency of which are listed again in Table 2. These include spondylosis, fractures of the dorsal vertebral processes, osteoarthritis, and, above all, impingement and overlapping spinous processes. The phenomenon known as “kissing spines” is considered to be a common major cause of poor performance as it is closely associated with thoracolumbar pain, which can have a significant impact on the horse’s welfare and performance. Secondary EBDs are much rarer and are triggered by the lameness of the limbs (Table 2) [48,49].

#### 3.1.4. EIPH: Exercise-Induced Pulmonary Hemorrhage

The prevalence of EIPH in polo horses ranges from 11% to 29.7% in various studies. In two studies, horses were examined for exercise-induced pulmonary hemorrhage (EIPH) after intense exercise [50,51].

In one study, neither blood on the nostrils nor the below-average performance of horses was observed [50]. In another study, the prevalence of EIPH was 29.7% (11 out of 37). Despite this significant percentage, the comparison between EIPH-positive and EIPH-negative horses showed no significant difference in tracheal cytology. It was found that polo horses are affected, to a relevant extent, by IAD and EIPH, so these respiratory diseases must be considered when assessing their health status. In comparison, the incidence of IAD in polo horses is the same as in event horses but not as high as in racehorses [50].

A study from 2003 showed that 46% of the animals examined were EIPH-positive, based on a careful classification of the animals according to the polo seasons played and sex, as well as endoscopic examinations of the airways. However, the statistical analyses in the study indicated that neither the number of polo seasons played nor the sex of the horse had a significant influence on the occurrence of EIPH [52].

#### 3.1.5. Myositis und Rhabdomyolysis

Rhabdomyolysis is another important stress-induced disease in polo horses. A survey of 423 polo horses revealed an average incidence of rhabdomyolysis of 7.3%, which caused considerable sporting and financial damage [53].

Creatine kinase (CK) plays a major role in myositis and rhabdomyolysis in the polo horse, as it is a biomarker for muscle damage. Of great scientific interest is the relationship between serum CK activity, the clinical condition during exercise, and the occurrence of musculoskeletal injuries. According to a study from 2008, the resting value of CK activity is 255 ± 9 iu/L [54]. Studies have shown that CK activity in plasma increases by 35% after intensive training, which may be related to muscle damage or a temporary increase in the permeability of the muscle fiber membrane [55].

In this context, it is interesting to note that although previous studies suggest that female polo horses are more susceptible to the disease, polo mares were not significantly more affected than geldings. Genetic factors may play a role. When examining the genome of various horses with rhabdomyolysis due to polysaccharide storage myopathy, a mutation in the glycogen synthetase 1 gene was found in one polo horse. Such genetic markers are an important cause of exertional rhabdomyolysis but are not responsible for all forms of polysaccharide storage myopathy (PSSM1), but they are crucial for the evaluation and management of the affected horses [56].

The management of polo horses differs from that of thoroughbreds and is less likely to be a risk factor for certain diseases such as rhabdomyolysis. Although this disease can occur in horses up to fifteen years of age, two-year-old horses are at higher risk [57,58].

It was found that supplementation with the ADE vitamin complex in a dose of 1 mL/50 kg of live weight (270,000 IU of vitamin A, 80,000 IU of vitamin D, and 80 IU of vitamin E/mL) via intramuscular injection can make muscle metabolism more efficient during exercise, suggesting a possible therapeutic approach [59]. After the game, the polo horses in the treated group had higher AST, lactate, and glucose levels than the horses in the control group. This indicates that the treated animals used their energy more efficiently. Before the game, the animals in the treated group also had higher GSH levels and lower lactate levels, indicating a better anti-oxidant supply [59].

#### 3.1.6. Equine Infectious Anemia (EIA)

Equine infectious anemia (EIA) is a chronic and potentially fatal viral disease that attacks the immune system of the affected animals [60]. It is of high veterinary relevance and has been reported in polo horses, including a case in 2017 in Switzerland’s Aargau canton. The eradication of the EIA virus is proving to be extremely difficult due to the long incubation period and high horse traffic [61]. In recent years, equine infectious anemia (EIA) has also become a concern in Germany due to its geographical proximity [14]. In a study conducted in Nigeria, 84 sera were collected from racehorses and polo horses in various local regions and tested for antibodies against the EIA virus using an indirect ELISA test. Positive results were found in 1.2% of the samples tested. It is worth noting that the positive horses were adults and had no fever or symptoms associated with EIA [62]. The Hurlingham Polo Association (HPA) focuses on preventing infectious equine diseases such as druse, dermatophytosis, equine herpesvirus (EHV), and equine influenza [7].

#### 3.1.7. Heart Disease

The first scientific report on the prevalence of heart murmurs in polo horses was conducted in 2019. The study showed a significant incidence of tricuspid and aortic murmurs [63]. Pathological heart murmurs are primarily attributable to the thickening of the valves as a consequence of the aging process, prolapse of the aortic valve, congenital malformations, diseases of the aortic root, and infective endocarditis of the heart [64]. However, the data were collected by means of detailed chest auscultation and thorough echocardiographic examination and did not reveal any striking differences compared to other horse populations [63].

### 3.2. Animal Welfare in Polo

In Germany, the Animal Welfare Act (Tierschutzgesetz) is the primary guideline for animal welfare. The law is based on Article 20a of the Basic Law. The original version dates from 24 November 1933, and has been continuously amended since then. Compliance with this act is not only a moral obligation but also a legal requirement in polo [65]. The health of the horses is paramount in the sport. The Hurlingham Polo Association has drawn up clear rules that define the health of the horses as a crucial prerequisite for the game. These rules, therefore, stipulate certain conditions under which a polo horse may not be used. These include cases where the horse is blind in one eye, has an open tracheotomy, has been enervated (whether by chemical or surgical means), or is temporarily desensitized. Similarly, polo horses that display unsafe behavior, such as kicking or biting, or show signs of infectious diseases, lameness, spur injuries, or bleeding, should not be used in play.

There are guidelines for extreme heat conditions when temperatures are very high, known as the “Guidelines for Extreme Heat”. These rules, along with the vaccination regulations of the Hurlingham Polo Association (HPA) and the German Polo Association (DPV), are important for protecting the health of horses. The associations also provide information on current hygiene practices and how to recognize and prevent the spread of diseases in horses. This is crucial for ensuring that horses remain healthy in the sport [5].

The Fédération Equestre Internationale (FEI) has recently strengthened its commitment to equine welfare with updated guidelines and protocols designed to ensure the comprehensive care and treatment of competing horses. Equine welfare has become an increasingly important issue in the equestrian world due to growing public concern and the concept of the Social Licence to Operate (SLO) committee.

The FEI document emphasizes a holistic approach to equine welfare, including appropriate nutrition, medical care, humane training methods, and ensuring psychological well-being [66,67].

## 4. Discussion

Polo horse health is a multifaceted issue that encompasses internal diseases, orthopedic injuries, and ethical aspects. Legal regulations and rules established by various organizations serve as a framework to ensure the protection of horse health in polo. However, the welfare of polo horses also depends on the experience and sensitivity of the players. It is essential for players to recognize the stress limits of their horses and to respond appropriately. They must also consider external factors like temperature and ground conditions.

Polo horses require demanding athletic skills such as speed, agility, and endurance. During the game, tendons and joints are at constant risk, with the most common orthopedic problem being injury to the superficial digital flexor tendon. Joint diseases, such as osteoarthritis, are particularly common among polo horses and can affect their longevity. Polo horses and racehorses share similar internal diseases. Comprehensive management strategies are required to mitigate the impact of exercise-induced pulmonary hemorrhage (EIPH) and rhabdomyolysis on equine health.

Traumatic injuries to horses in polo, particularly from balls and mallets, raise animal welfare and ethical concerns. To address these concerns, preventative measures should be taken, and prompt medical action should be initiated in the event of injury. It is also essential to comply with the regulations established by organizations such as the Hurlingham Polo Association to protect the welfare of polo horses. The regulations set out the conditions under which polo horses should not be used and emphasize the importance of taking proactive measures to protect the health of the horses. Guidelines for extreme heat conditions, vaccination protocols, and hygiene practices help maintain the health of the horses and prevent the spread of disease. Veterinarians play a crucial role in treating sick horses and preventing injuries. For example, organizations and players receive advice on optimizing ground adjustments and shoeing strategies. The topic of soil composition in polo fields is a broad one. Despite its importance, it has not been well studied in the literature.

Responsible management practices, compliance with regulations, and ongoing efforts to improve player awareness are crucial to protecting the welfare of polo horses. To promote the health and welfare of polo horses, it is essential for associations, players, organizers, and veterinarians to cooperate.

In addition to the ethical considerations pertaining to animal welfare and the responsible treatment of polo horses, there are other important ethical aspects to polo. It is of paramount importance to consider the environmental impact of polo, particularly with regard to land use and resource consumption. It is similarly important to promote education and raise awareness and information, which is the responsibility of all those involved. In this context, transparency and openness also play a decisive role in relation to the treatment of horses, playing conditions, and sporting performance. Moreover, exemplifying fair play as an ethical principle is of central importance for the integrity of polo.

## Figures and Tables

**Table 1 animals-14-01735-t001:** Means and standard deviations of significant biochemical parameters of thoroughbred horses in training and the athletic participants of official polo matches [8].

Parameter	Thoroughbred Horses in Training (Mean ± SD ^1^)	Athletic Participants of Official Polo Matches (Mean ± SD ^1^)
Lactate (mmol/L)	0.71 ± 0.15	0.59 ± 0.11
Alkaline phosphatase (U/L)	146.30 ± 103.40	330.40 ± 94.30
Gamma-glutamyl transferase (U/L)	232.30 ± 49.60	174.9 ± 48.30
Urea (mmol/L)	6.82 ± 2.93	5.39 ± 1.57
Creatinine (µmol/L)	3380 ± 930	3958 ± 841
Phosphorus (mmol/L)	0.33 ± 0.12	0.41 ± 0.05
Potassium (mmol/L)	4.50 ± 0.60	5.33 ± 0.83

^1^ SD: standard deviation.

**Table 2 animals-14-01735-t002:** Types of primary and secondary equine back disorders shown as percentage distributions according to a study from 2019 [48].

Equine Back Disorder	Type of Vertebral Lesion	Distribution
Primary EBDs (92.27%)	Overriding DSPs (kissing spines)	80.65%
	Spondylosis	9.68%
	Osteoarthritis	6.45%
	Fractured dorsal spinal processes	3.22%
Secondary EBDs (7.73%)	Forelimbs	(2.21%)
	Hindlimbs	(5.52%)

## Data Availability

The data presented in this study are available on request from the corresponding author. The data are not publicly available due to privacy restrictions.

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
