# Peer review of "Health of Polo Horses"

_animals, 2024, doi:10.3390/ani14121735_

Round 1
Reviewer 1 Report
Comments and Suggestions for Authors
Dear Authors, please correct the marked sentences!

Author Response
"Please see the attachment."

Reviewer 2 Report
Comments and Suggestions for Authors
At least a paragraph on comparative injury and pulmonary hemorrhage rate in horses used for other sports, racing, of course and reining 36 -46 Instructions to authors included in introduction
124 mention indoor polo
152 but eventing is not polo. The uses are very different.
156 "breaking in" do you mean training; these are not wild mustangs
163 great not high
178 because the field is smaller in indoor polo so horse have to run less?
162 but endurance horses are not ridden at high speed. Are you discussing another discipline?
211 inverse correlation?
223 Half clinically lame? Need a reference for this devasting statement.
245 omit "which are prone to joint disease" unless you mean only those ponies already at risk, independent of use as polo ponies.
257 what is the % in the recent publication
264 what is polocrosse. This reviewer is not aware of the sport so other readers may also be unaware
266 bone spurs on which bones?
286 move definition of kissing spines to preceeding sentence overlapping of spinous processes, kissing spines,
302multiparous -you don't mean polo ponies who have had several foals. What do you mean? Horses used for other purpose such as dressage jumping, endurance.
326 more important than what?
332 contagious? It is insect-borne, but there isn't horse to horse spread is there
342 significant ( what was compared )?
363 extreme environmental or weather conditions
373What are the ethical aspects- only the diseases and injuries have been mentioned
376 has pitch been mentioned before?
381 longevity as polo mounts ( presumably not their life span)
Comments on the Quality of English Language.
Author Response
"Please see the attachment."

Reviewer 3 Report
Comments and Suggestions for Authors
The manuscript "Health of Polo Horses" focus on several injuries which polo horses are exposed to. The revision contents useful information that may contribute to protect the health of horses in polo and minimize injuries. In my oppinion, that question is address in the manuscript. Maybe the authors could develop more what kind of accidents Polo horses are prone during competitions, but probabbly there are not sufficient information in the literature... Considering that is a revision manuscript, the authors are limited by the available information in the literature, But try to give that information.
The authors corrected the title of the manuscript to better reflect its content.
The simple summary is adequate.
The Abstract should be corrected to elucidates the content of the manuscript. I suggest to remove the sentence: “The abstract presents an overview of animal health in polo through a systematic 16 analysis of relevant literature.”
The Introduction must be corrected.
Please remove the sentences between lines 38-45.
The authors refer 3 times the aims of the revision in different sentences. I suggest to review the paragraph and use a better structure to the information in the introduction.
Line 75 – delete “I”
The results section is sound.
Line 310 – correct to: Rhabdomyolysis
The discussion seems correct.
The authors could develop more the updated treatments to Polo horse's injuries, even thought those treatments are not specific for Polo horses.Regarding diagnosis, the authors could give more particular informations regarding the type of lesions. For exemple, they could give more informations regarding SFDT like localization fo the lesions (proximal, distal) the extensions of lesions, etc and compare that to lesiond occuring is other sports. The authors could give some ilustrations of tipical injuries observed in Polo horses...
Author Response
"Please see the attachment."

Reviewer 4 Report
Comments and Suggestions for Authors
Thank you for the opportunity to review this paper. It's great to see more work being done in Polo, and putting the horses first. Given the style of the review conducted and the nature of the topic(s) within it, I think you need to consider including several figures to support key areas of the text; this should include but is not limited to, the review protocol and results. Tables should also be used to group common texts. Please find a line by line review, and some general comments per section, below.
Abstract:
Introduction:
Lines 38-45 need deleting, as these are part of the submission template.
51 - you may wish to be broader and amend diseases to 'veterinary factors', as you discuss injury etc.
54 - interesting to note 'correlations', I'm keen to see how you've performed this given potential differences in data reporting between studies
57 - I'd consider deleting solid, as this is subjective
Your introduction is very short, and doesn't really introduce Polo from the horses' perspective. What are the requirements of the game, how does this differ between levels of play? How might environmental factors impact this? Are there reviews of other performance horse sports that are similar or different to Polo? How did you develop the concern for the veterinary factors within the sport? For some general basics of the sport, papers by Best are a good starting point. You can then build your veterinary factors around this. Noleto's paper on the biochemical responses to polo, and Williams' heart rate paper will also support this description of the sport.
Methods:
65-66 - can you be more specific about your search strategy - perhaps use a table?
66-68: How many titles did you retrieve? How many were excluded, and on what basis? Did you only include texts relevant to Polo, or did you include texts where Polo ponies made up some of the sample? Did you exclude literature that assessed Polocrosse, too?
68 - you might need to change this sentence, as it suggests Polo is an independent variable i.e. you had healthy horses and they've been introduced to Polo and tracked extensively, possibly compared to a non-Polo group. This isn't the typical design for these studies.
71 - did you have anyone to double check or decide on your inclusion of studies, if authors disagreed on their suitability?
75-80 - I think the inclusion of grey literature here is important, given the context it will provide to the academic discussions and limited scientific literature. Your critique is valid; but I'd like to know more about how you searched Google, and organisation websites (which ones, reference etc.) e.g. did you use the same Boolean operators or structure your searching differently?
Results
Please include a PRISMA diagram or similar to depict your search strategy and results. You might want to include a table for each variable/category and outline key references and findings from that section. This is common in systematic reviews across a range of fields, but clearly shows the reviewers that you've covered a breadth of literature in each area.
You might also want to change your results writing style to simply report on the number of texts and typical results in each area. Use tables to support this. You can then do a much better job of exploring each topic and groupings of papers in your discussion
84 - 153 - each of these sections should likely sit in your introduction. They provide important background, and don't necessarily align to the aim of your review but contextualise it. From a veterinary perspective, I think some results/papers related to breeding/cloning would be appropriate for your results, as they feed into reproductive health and practice, as well as having an indirect impact upon Polo performance and welfare. e.g. the ethical and long term consequences of cloning have not been considered, or properly articulated. You may wish to include this in your discussion, too.
160/162 - this statement is misleading. I think you need to properly define the principles of training before making statements such as this. There's clearly adaptation to training stressors that would take place over a six year time period, so adjustments in absolute intensity are likely required but relative intensity may remain the same in the individual horse(s), if provided adequate recovery?
163/165 - I think you can expand this sentence. It is well referenced, and each point is valuable, but as a reader I want to know more about each statement made.
167/170 - are the variations due to the condition of each horse, or different physiological responses to each type of training session? I think more detail is required here.
171 - muscle strengthening suggests a hypertrophic response is desired i.e. an increase in muscle mass, but you go on to talk about increasing oxidative capacity. This is more aerobic in nature and is driven either by shifts in muscle fibre type, or an increase in mitochondrial density and protein activity. These are distinct training types and responses. Strength is certainly important, but not for the reasons stated. Please reconsider.
175 - can you expand a little more here? How and why does glycolytic exercise increase the risk of rhabdomyolysis in Polo ponies? Is equine muscle more biased to glycolytic activity - consider some of the classic literature in equine physiology here e.g.
Armstrong, R.B.; Essen-Gustavsson, B.; Hoppeler, H.; Jones, J.H.; Kayar, S.R.; Laughlin, M.H.; Lindholm, A.; Longworth, K.E.; Taylor, C.R.; Weibel, E.R. O2 Delivery at VO2max and Oxidative Capacity in Muscles of Standardbred Horses. J. Appl. Physiol. 1992, 73, 2274–2282, doi:10.1152/jappl.1992.73.6.2274.
Poole, D.C.; Erickson, H.H. Highly Athletic Terrestrial Mammals: Horses and Dogs. Comprehensive Physiology 2011, 1, 1–37, doi:10.1002/cphy.c091001.
167/184 - consider breaking this up into several paragraphs and discussing each theme in more detail e.g. physiological responses to training, glycolytic metabolism, aerobic and strength adaptations as a result of training
186/204 - consider placing this as either the first or last section in your results as it describes the 'end product'. This section also misses key papers that explore the differences between Open and Womens Polo, and the variability in spatiotemporal characteristics. Heart rate literature is missing too.
Best, R.; Standing, R. All Things Being Equal: Spatiotemporal Differences between Open and Women’s 16-Goal Polo. International Journal of Performance Analysis in Sport 2019, 19, 919–929, doi:10.1080/24748668.2019.1681790.
Best, R. Within and Between-Tournament Variability in Equestrian Polo. J Equine Vet Sci 2022, 119, 104144, doi:10.1016/j.jevs.2022.104144.
Best, R.; Standing, R. External Loading Characteristics of Polo Ponies and Corresponding Player Heart Rate Responses in 16-Goal Polo. J Equine Vet Sci 2021, 98, 103368, doi:10.1016/j.jevs.2020.103368.
Williams, J.M.; Fiander, A. The Impact of Full vs. Half Chukka Playing Strategies on Recovery in Low Goal Polo Ponies. Comparative Exercise Physiology 2014, 10, 139–145, doi:10.3920/cep140004.
205 - I think you need to redefine this section; disease typically refers to conditions that onset gradually and have a known cause. Your first three, arguably four, subsections are all concerned with injury, which are acute and whilst most have known causes may be due to random chance/bad luck. I have not reviewed these in detail at this stage, due to the extent of recommended reviews thus far.
352 - you might want to include comment on social license to operate here, and how this is being led by the FEI with the ISES in other disciplines. Recent rule changes by the HPA in 2024, should also be included e.g. maximum spur length, hitting into the horse rule, excessive use of the whip etc.
370/399. This section behaves more like a conclusion. I've not reviewed extensively, but it appears to include all key aspects mentioned in the sections above.
General feedback - I think this is an important paper, with the potential to be well cited. I for instance would cite it in most papers that we plan to write, if sufficient changes are made and the information included increases in depth and breadth. You've got a unique paper here that potentially showcases Polo ponies as great athletes, but as with all great athlete they are susceptible to injury either through poor management of training load, other health factors or due to random chance. Social license to operate and welfare are outlined, and the role of governing bodies in this is acknowledged but more updated evidence is required.
Again, I'll reiterate the need to support the text with tables and figures. These are easily shared in presentations, lectures and on social media and serve to increase the potential visibility of your work. I appreciate there are plentiful revisions stated above, but you are missing key papers and themes, and the paper requires some further organisation.
Comments on the Quality of English LanguageThe English throughout is largely ok, but requires some revision to reflect the strength of arguments presented, and the weight of evidence.
I'd like to see authors tend to the major revisions required first, and English be revised subsequently
Author Response
"Please see the attachment."

Round 2
Reviewer 1 Report
Comments and Suggestions for Authors See Table 1
Reviewer 4 Report
Comments and Suggestions for Authors
Thank you for a revised version of the manuscript. I'd like to commend the authors on the revisions made thus far, but have a few more suggestions prior to recommending acceptance. Apologies if I get line numbers wrong, they have changed between drafts and seem to skip from 88 - 223?
86 - indoor/arena polo is a valid comment, if you want further comparisons I think the Pritchard paper likely has what you need - , , . Cardiorespiratory responses reach vigorous-intensity levels during simulated gameplay of arena polo. Transl Sports Med. 2020; 3: 250–255. https://doi.org/10.1002/tsm2.128
87 - chukka length is 7:30 of 'ball-in-play' time. This doesn't account for stoppages due to penalties, which increase the length of 'real' chukka time i.e. the time players spend on horses. This distinction is important, as it may increase risk for injury and/or afford horses a recovery opportunity or for players to change horses. Some data on real chukka time in the following:
Best, R.; Standing, R. External Loading Characteristics of Polo Ponies and Corresponding Player Heart Rate Responses in 16-Goal Polo. J Equine Vet Sci 2021, 98, 103368, doi:10.1016/j.jevs.2020.103368.
Best, R. Within and Between-Tournament Variability in Equestrian Polo. J Equine Vet Sci 2022, 119, 104144, doi:10.1016/j.jevs.2022.104144.
I wonder if you need to consider either the order of your introduction, or pushing some of the historical and pony comments down into your results. I think the introduction down to line 234 is really well-written and this would be an appropriate ending point for your introduction. 234/448 may make for great discussion paragraphs as all provide additional and valuable context. This is obviously up to you as authors, but I am really unsure of the fit of these paragraphs within the introduction at present/within their current position.
Your methods are much improved, and a lot clearer. The acknowledgement of bias but 'richness' adds value to the reader.
472 - check the wording around training starting
481 - consider using the Best variability reference here, to support your statement.
482 - what do you mean by pauses? Stoppages in play due to umpiring decisions or similar?
483 - consider your ref 3 here, too as it actually measures speed
556-567 - I really like the additional detail put in here, but these statements require support from references. I believe I provided some for consideration in previous review?
569/570 - How was the lactate assessed? Was it via blood or muscle biopsy? I'd be cautious around using the term acidosis if hydrogen concentration isn't assessed or accounted for. Just using Lactate may be more consistent with your other text and table. Also check plasma and blood wording and ensure this aligns with references - fine if correctly matched to refs.
577 - consider lengthening Game to game analysis or game demands
649 - you might want to include a table or figure adapted from some references here, to support the sentence you've written citing ref [29]. Other references will also support this statement. Make it clear to the reader what these values are, and how they may change by level/sex played. Expand text a little, too.
649 - start new paragraph on warm up
659 - 'fitness' might be better replaced by metabolism? Consider linking to other tables, or specific HR values in papers you've cited previously
676 - starts? Is this term borrowed from the racing literature? How does this compare to other disciplines?
685 - farrier would also be responsible for shoeing, perhaps moreso than a vet. Likely limited data on different types of grass, but you may find some interesting comparisons to racing here with natural and artifical surfaces, or differences in 'going' and the prevalence and severity or site of injury.
689 - consider stating sample size in brackets. Be careful with your wording here as it sounds like half of all polo ponies are clinically lame. Interesting that horses were trotted too, when Polo is played predominantly at a canter/gallop so lameness may dissipate with increasing speed or is in fact an (functional) asymmetry as opposed to lameness
698 - add digital if required, too
797-816 - this section covers quite a range of trauma, you might want to define trauma briefly at the start, before progressing into each type e.g. acute, chronic
807 - ensure consistent spelling of Polocrosse
816 - check correlation sentence as this is obvious, I think you can just write a combined sentence which covers age, playing time and oral lesions/injury - again be clear about type of trauma you're referring to.
822 - range of breeds here, are there references to support? Is this a product of being a sportshorse i.e. if ridden for sport performance, back injury is more likely due to increase volume and intensity of ridden work compared to a hack/pleasure ride?
911 - great table, but percentages in brackets aren't clear? Has this been adapted from another source(s) - if so cite accordingly
930/1026 - some of your other metabolites e.g. aminotransferase from previous paragraphs are also markers of muscle and kidney damage. It'd be good to bring these back in here, to better contextualise the acute exercise demands. This also highlights that adaptation to exercise is a biologically stressful and active process, thus studies should employ repeated measures designs to assess the consistency of biological/haematological pertubations in response to polo/ridden exercise
1022 - sex predisposition? Can you be more specific?
1025 - what is meant by efficient here? Efficiency within a muscle would suggest to me that there is a greater yield of ATP per molecule of substrate being oxidised? Do the A and E provide an anti-oxidant effect? What is the measure and mechanism associated with efficiency in this context?
1038/1039 - you might want to move your sentence re Germany to be next to your Switzerland sentence, given their geographical proximity.
1038 - presuming that the HPA does this through use of the Equine passport system?
In your welfare section you might wish to cite the recent FEI document regarding welfare in response to social license to operate. This is a concern in Polo, too and the sport would be advised to consider adopting relevant parts of the FEI model to ensure the horse's welfare is managed holistically
Overall the manuscript is much improved, I think there's still some further revision to be recommended as per the above. A consistent theme in the writing is either not finishing a thought, and not citing consistently and frequently. It's ok and encouraged to cite the same source(s) repeatedly where they add value or provide a reference point; at present you often wait until the end of a sentence, or even paragraph to reference.
Comments on the Quality of English LanguageEnglish language quality is much improved.
